# NHC-catalyzed atropoenantioselective synthesis of axially chiral biaryl amino alcohols via a cooperative strategy

Gongming Yang[1], Donghui Guo[1], Di Meng[1] & Jian Wang[1]

Axially chiral biaryl amino-alcohols play a pivotal role in organic synthesis and drug discovery. However, only a very few enantioselective methods have been reported to synthesize chiral biaryl amino-alcohols. Therefore, the rapid enantioselective construction of optically active biaryl amino-alcohols still remains a formidable challenge. Here we report an N-heterocyclic carbene (NHC)-catalyzed atropoenantioselective acylation of biphenols triggered by a cooperative strategy consisting of desymmetrization followed by kinetic resolution. This protocol features broad substrate scope and good functional group tolerance, and allows for a rapid construction of axially chiral biaryl amino-alcohols in good to high yields and with excellent enantioselectivities. Furthermore, the structurally diverse axially chiral biaryl amino-alcohol derivatives provide multiple possibilities for chemists to develop catalysts or ligands for different chemical transformations.

---

[1] School of Pharmaceutical Sciences, Collaborative Innovation Center for Diagnosis and Treatment of Infectious Diseases Key Laboratory of Bioorganic Phosphorous Chemistry and Chemical Biology (Ministry of Education), Tsinghua University, Beijing 100084, China. Correspondence and requests for materials should be addressed to J.W. (email: wangjian2012@tsinghua.edu.cn)

Axially chiral biaryls[1–3] have widely applied in many areas, including material science[4,5] and drug discovery[6]. In addition, chiral biaryls have often played as ligands[7] or catalysts[8] in the development of enantioselective catalytic transformations. During the past two decades, BINOL and BINAM have proven to be representative examples in this category[9–14]. Afterward, NOBIN (Fig. 1) gradually grows up to the next privileged scaffold[15] with numerous important applications, due to its perfect enantio-control and prominent bioactivity (Fig. 1, (R)-Streptonigrin, an antitumor agent).

In sharp contrast to the asymmetric preparation of BINOL or BINAM[16–19], to date, only a very few enantioselective methods have been reported to synthesize NOBIN-type biaryl amino alcohols[20–23]. In 1992, Kocovsky[24] and coworkers reported the asymmetric synthesis of NOBIN via oxidative of 2-naphthol with 2-naphylamine. However, the use of large excess amount of chiral auxiliary and required multistep crystallization sometimes restrict its further application. To face this issue, Tan's group[25] uncovered a phosphoric acid-catalyzed cross-coupling of 2-naphthylamines with iminoquinones, affording chiral biaryl amino alcohols in a concise and catalytic pattern. Shortly after, optical resolution rapidly developed into the next attractive method to isolate NOBIN enantiomers by leveraging the power of diastereoisomeric chiral salt formation, but was confined to unstable reproducibility in practical[26,27]. Getting the NOBIN derivatives through a direct transformation from chiral raw materials (e.g., BINOL[28] or BINAM[29]) has also become an interesting way. Regrettably, this protocol is mostly applied to construct chiral binaphthyl-type amino-alcohols. Recently, kinetic resolution is recognized as an impresive technology to produce such biaryl structures. For example, the groups of Maruoka[30] and Zhao[31] reported a phase-transfer- or N-heterocyclic carbene-catalyzed asymmetric kinetic resolution to prepare enantioenriched chiral biaryl amino alcohols, independently (Fig. 1). However, no more than 50% theoretical yields inevitably affect the application of this approach. Overall, rapid synthesis of axially chiral biaryl amino-alcohols in a highly atropoenantioselective fashion is still in its infancy

Our group is interested in exploring carbene catalysis for the rapid assembling of axially chiral molecules. To date, we have successfully reported the N-heterocyclic carbene-catalyzed atropoenantioselective [3 + 3] annulation and kinetic resolution of anilides, affording valuable chiral α-pyrone-aryls and iso-indolinones, respectively[32,33]. Despite aforementioned achievements, the unsolved challenges and the continuously growing demands of atropoenantiomers still drive us to develop more efficient and revolutionary protocols. We herein report a carbene-catalyzed[34–41] atroposelective synthesis of axially chiral biaryl amino-alcohols via a cascade strategy of desymmetrization followed by kinetic resolution[42] (Fig. 1). First, this approach can deliver nonclassical NOBIN derivatives (e.g., biphenyl- or phenyl-naphthyl-type amino-alcohols) in a high chemical yield. From the aspect of structural diversity, non-classical NOBIN-type derivatives will offer more possibilities for the exploration of new chiral catalysts or ligands. Second, in contrast to an independent desymmetrization or kinetic resolution method, the cooperation of desymmetrization[43–48] with kinetic resolution[49–53] has certain superiority in the control of enantioselectivity.

## Results

**Reaction optimization.** We commenced our study by using biphenols (**1a–c**) as the model prochiral substrates, aldehyde (**2a**) as acylation reagent[54–62] and DQ as oxidant[63,64]. Key results of reaction optimization are briefly summarized in Table 1. Building upon the indanol-derived triazolium scaffold, precatalysts with

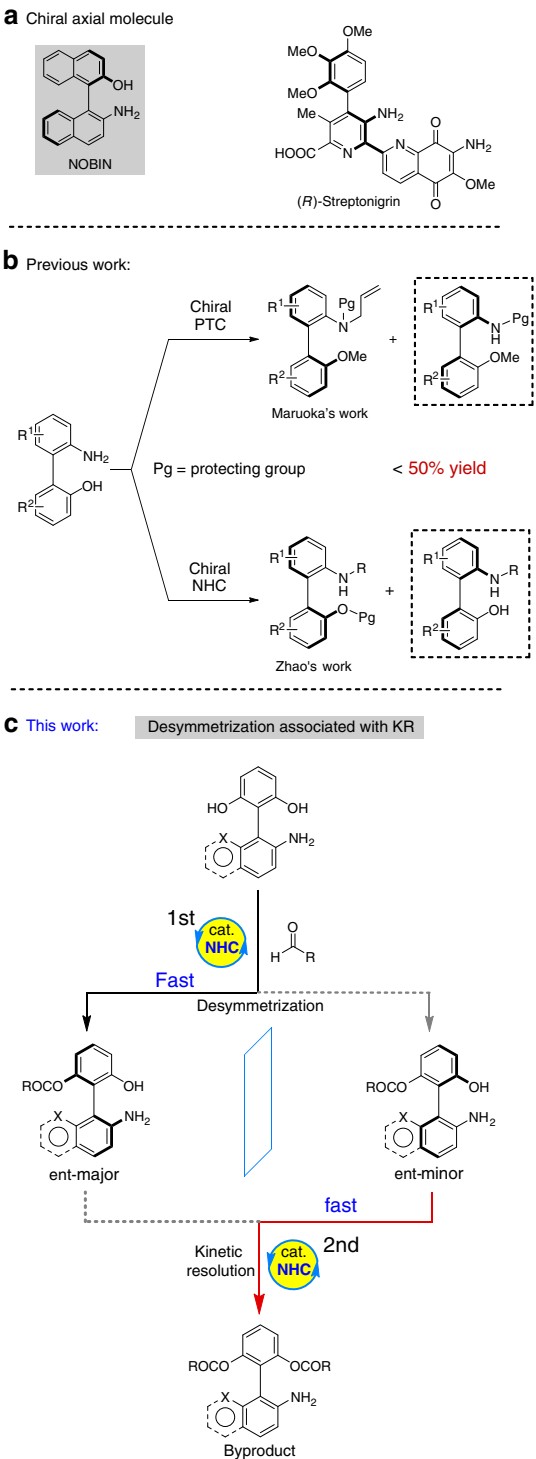

**a** Chiral axial molecule

NOBIN

(R)-Streptonigrin

**b** Previous work:

Chiral PTC — Maruoka's work — Pg = protecting group — < 50% yield

Chiral NHC — Zhao's work

**c** This work: Desymmetrization associated with KR

1st cat. NHC — Fast — Desymmetrization

ent-major — ent-minor

fast — Kinetic resolution — 2nd cat. NHC

Byproduct

**Fig. 1** Representative molecules and synthetic protocols. **a** Two representative axially chiral molecules. **b** Asymmetric kinetic resolution of achiral biaryl amino alcohols. **c** Our synthetic proposal via a NHC-catalyzed atroposelective synthesis of axially chiral biaryl amino-alcohols via a cascade strategy of desymmetrization followed by kinetic resolution

N-2,4,6-(Cl)$_3$C$_6$H$_2$[65], or N-C$_6$F$_5$[66], substituents (Table 1, **C2** and **C3**), derived from α-amino acids, were tested but exhibited low conversions and enantioselectivities (Table 1, entries 4 and 5). Interestingly, precatalyst **C1** with N-2,4,6-(Me)$_3$C$_6$H$_2$ (N-Mes)[67] substituent provided **3a** in 63% with 85% ee (Table 1, entry 1). If substrate **1b** or **1c** replaced **1a** (R = NO$_2$ or NH$_2$; More information about changing R groups, see Supplementary Table 2),

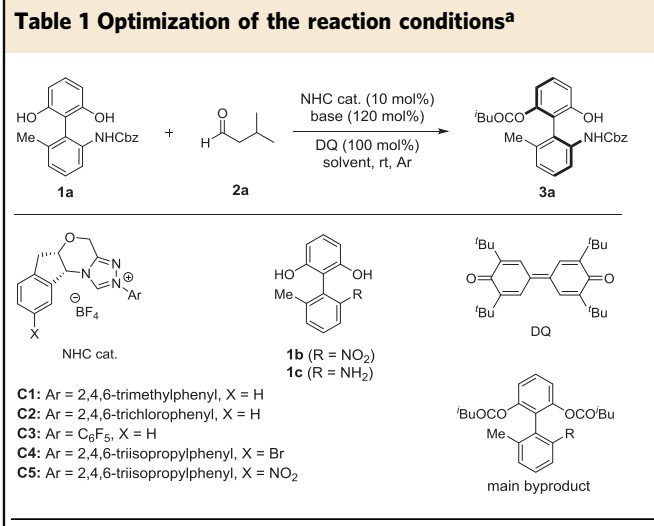

**Table 1 Optimization of the reaction conditions[a]**

| Entry | NHC cat. | Solvent | Base | Yield (%)[b] | ee (%)[c] |
|---|---|---|---|---|---|
| 1 | C1 | THF | $K_2CO_3$ | 63 | 85 |
| 2[d] | C1 | THF | $K_2CO_3$ | 71 | 50 |
| 3[e] | C1 | THF | $K_2CO_3$ | 15 | □− |
| 4 | C2 | THF | $K_2CO_3$ | 23 | 23 |
| 5 | C3 | THF | $K_2CO_3$ | < 5 | □− |
| 6 | C4 | THF | $K_2CO_3$ | 74 | 90 |
| 7 | C5 | THF | $K_2CO_3$ | 80 | 95 |
| 8 | C5 | toluene | $K_2CO_3$ | 90 | 96 |
| 9 | C5 | MeCN | $K_2CO_3$ | 78 | 84 |
| 10 | C5 | MTBE | $K_2CO_3$ | 80 | 98 |
| 11 | C5 | DCM | $K_2CO_3$ | 95 | 96 |
| 12 | C5 | DCM | $C_{s2}CO_3$ | 85 | 92 |
| 13 | C5 | DCM | $Et_3N$ | 76 | 97 |
| 14[f] | C5 | DCM | $K_2CO_3$ | 78 | 93 |
| 15[g] | C5 | DCM | $K_2CO_3$ | 92 | 99 |
| 16[h] | C5 | DCM | $K_2CO_3$ | 87 | 96 |

[a]Conditions: **1a** (0.1 mmol), **2a** (0.15 mmol), catalyst (10 mol %), base (0.12 mmol) and DQ (0.1 mmol), solvent (1.0 mL), room temperature, Ar, 2 h.
[b]Isolated yields after flash column chromatography
[c]Enantiomeric ratio (ee) determined via chiral-phase HPLC analysis
[d]**1b** replaced **1a**
[e]**1c** replaced **1a**
[f]**2a** (0.12 mmol), 15 h
[g]DQ (0.12 mmol), 12 h
[h]**C5** (5 mol %), DQ (0.12 mmol), 24 h.

**Fig. 2** Scope of aldehydes. Reaction conditions: a mixture of **1a** (0.10 mmol), **2** (0.15 mmol), $K_2CO_3$ (0.12 mmol), and DQ (0.12 mmol) in $CH_2Cl_2$ (1.0 mL) was stirred at room temperature under $N_2$ for 12–24 h

low ee and yield were observed (Table 1, entries 2 and 3). To our delight, further improved enantioselectivities were achieved with the N-2,4,6-($^iPr)_3C_6H_2$ substituted catalyst **C4** or **C5**[68] (Table 1, entries 6 and 7). Notably, the triazolium catalyst **C5**, which bears a strong electron-withdrawing group ($NO_2$) at the remote aryl position, afforded 95% ee and 80% yield (Table 1, entry 7). We then chose catalyst **C5**, substrate **1a** and **2a** for further optimization. After extensive screening of solvents, bases and catalyst loading, an ideal result was obtained by using 10 mol% of **C5** as catalyst, DCM as the solvent, and $K_2CO_3$ as the base (Table 1, entry 15). Meanwhile, nuclear magnetic resonance (NMR) spectrum confirmed that the main byproduct of this reaction was a bisadduct in which two hydroxyl groups were both acylated (see Supplementary Note 4).

**Substrate scope.** Having the optimal condition in hand, we turned our attention to the generality of aldehydes. As indicated in Fig. 2, a diverse set of aliphatic aldehydes underwent acylating reactions, affording their corresponding products in high yields with high to excellent ee values (Figs. 2 and 3b–i). Unambiguously, the steric effect of aldehydes has identified to be a critical factor for achieving high enantioselectivity. Aliphatic aldehydes bearing a steric bulky chain afforded a higher ee value (Figs. 2 and 3a, b, f, g). When aromatic aldehydes were used as substrates, reactions are generally messy, probably caused by the competitive benzoin reaction.

Encouraged by success with aliphatic aldehydes, we then planed to investigate the reactivity of biaryl-type biphenol substrates. N-Cbz protected compounds **4a–j** having electron-donating groups (Me, MeO) and/or electron-withdrawing groups (Cl, CN) on the lower ring (Fig. 3, phenyl ring B) generated the coresponding products in high yields (85–94%) with excellent enantioselectivities (96– >99% ee). Cyclohexane ring fused biphenyl substrate also performed well as expected (Fig. 3, **4l**, 91% yield and 96% ee). Pleasingly, the indole-based biaryl substrate also proceeded smoothly to afford product **4m** with a promising ee value. Gratifyingly, the phenyl-naphthalenyl-type biaryl substrate was also tolerated to deliver the anticipated structure **4n** with excellent enantioselectivity (Fig. 3, 99% ee). Incidentally, enantiomeric products can be achieved entirely through the enantiomer catalyst and similar reaction conditions (Fig. 3, **4o**).

**Mechanistic studies.** To verify the mechanism, two control experiments were conducted. As indicated in Fig. 1a, 4), ent-**3a** was obtained in 49% yield with 93% ee in the presence of DQ (0.6 equiv). On the basis of these data, we can conclude that desymmetrization is a key contributor for enantio-control ($V_{fast}/V_{slow} = 28:1$)[69]. Meanwhile, we also wonder whether the second acylation is a kinetic resolution process, eventually resulting in an improved enantioselectivity. To approve this hypothesis, the control experiment of (±)-**3a** with **2a** was designed and carried out, generating ent-**3a** in 47% yield with 76% ee (Fig. 1a, 4). This experiment result suggests that the conversion of **1a** to major enantiomer **3a** is much faster than the process between **1a** and minor enantiomer ($V_{fast}/V_{slow} = 5:1$)[69].

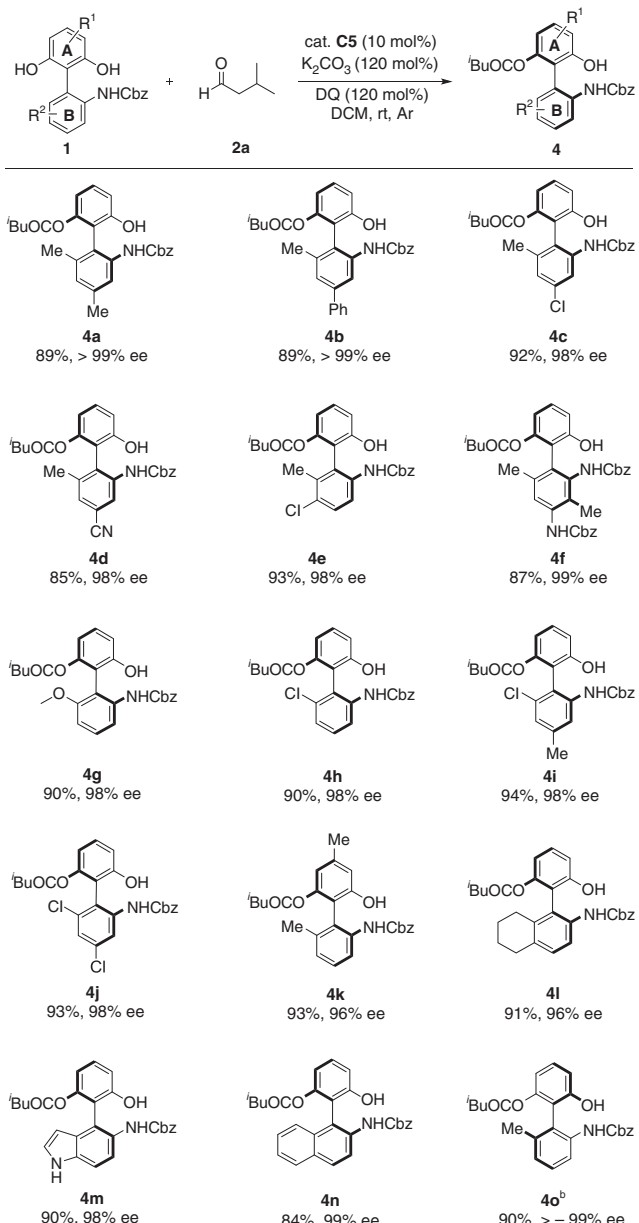

**Fig. 3** Scope of biaryl biphenols. Reaction conditions: a mixture of **4a–4o** (0.10 mmol), **2a** (0.15 mmol), $K_2CO_3$ (0.12 mmol), and DQ (0.12 mmol) in $CH_2Cl_2$ (1.0 mL) was stirred at room temperature under $N_2$ for 12–24 h. [b]ent-cat. **C5** was used

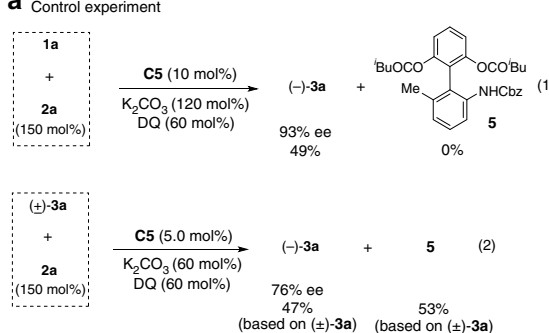

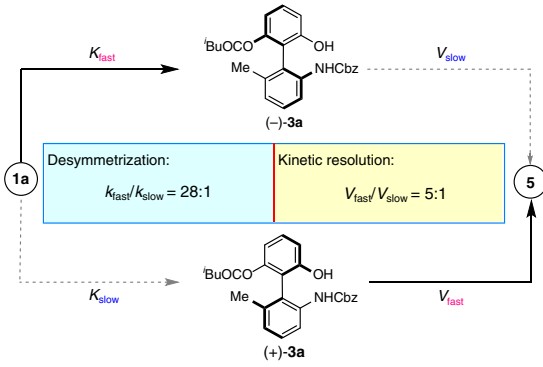

**Fig. 4** Postulated mechanistic pathways. **a** The control experiments (Eqs. (1) and (2)) show that the desymmetrization process is the main contributor to the observed ee of the product and the second acylation is a effective KR process that could improve the ee of (−)-**3a**. **b** The postulated mechanistic pathway to generate product (−)-**3a**

**Fig. 5** Synthetic transformations. Reaction conditions: (1) **4n**, TMSCHN₂, CHCl₃:MeOH/5:1, r.t., 24 h. (2) NaOMe, MeOH, r.t., 1.0 h. (3) Pd/C, H₂, MeOH, r.t., 3.0 h

determined by X-ray single crystal analysis (See Supplementary Fig. 1), and other structures were assigned by analogy.

## Discussion

In conclusion, we have developed an atropoenantioselective NHC-catalyzed acylation for the preparation of axially chiral biaryl amino-alcohols. The cascade strategy of desymmetrization followed by kinetic resolution could efficiently deliver axially chiral biaryl amino-alcohols with high to excellent ee values (up to >99% ee). Further studies on the exploration of other substrates and applications are ongoing projects in our laboratory.

**Synthetic transformations and applications**. We anticipated that the biaryl amino-alcohol **7** (Fig. 5), prepared from **4n** via a N-Cbz protected intermediate **6**, could be directly utilized as a chiral catalyst in asymmetric alkylation reaction. As highlighted in Fig. 6, compound **7** successfully catalyzed the asymmetric alkylation of Ni-complex **8** with alkyl bromide **9–11** to generate complex **12–14** with promising er values. After a subsequent deprotection, **12–14** could efficiently transfer to valuable chiral α-amino acids[70]. To further expand synthetic utility, we conducted a Ru-catalyzed asymmetric reduction of ketone **18** by using chiral biaryl amino-alcohol derivatives **15–17** as ligands (for preparation of **15–17**, see Supplementary Note 5). As indicated in Fig. 6, ligands **15–17** led to the corresponding product **19** in suggestable er values. In addition, a gram-scale synthesis (3.4 mmol) carried out under standard conditions afforded optically pure **3a** in a pleasant result (Fig. 7, 89% yield, 99% ee). The absolute configuration of derivative **6** was

## Methods

**Synthesis of racemic ¾**. In a glovebox, a flame-dried Schlenk reaction tube equipped with a magnetic stir bar, were added racemic NHC precatalyst **C13** (0.01 mmol,), $K_2CO_3$ (16.6 mg, 0.12 mmol), oxidant DQ (49.0 mg, 0.12 mmol), **1** (0.10 mmol), **2** (0.15 mmol), and freshly distilled $CH_2Cl_2$ (1.0 mL). The reaction mixture was stirred at room temperature for 12 h. The mixture was then filtered through a pad of Celite washed with $CH_2Cl_2$. After solvent was evaporated, the residue was purified by flash column chromatography to afford the racemic product **3/4**.

**a** Asymmetric alkylation

| | **9–11** | Product | er | Yield |
|---|---|---|---|---|
| ●— | R—Bn | 12 | 94:6 | 95% |
| ●— | naphthyl | 13 | 86:14 | 94% |
| ●— | allyl | 14 | 90:10 | 91% |

**b** Asymmetric reduction

Ligand 15
89%, 83:17 er

Ligand 16
91%, 59:41 er

Ligand 17
92%, 70:30 er

**Fig. 6** Synthetic applications. **a** Use of **7** as a chiral catalyst. **b** Utility of **15–17** as chiral ligands

**Fig. 7** Gram-scale synthesis. Reaction conditions: a mixture of **1a** (3.4 mmol, 1.19 g), **2a** (5.1 mmol, 0.55 mL), in $CH_2Cl_2$ (34.0 mL) was stirred at room temperature for 15 h

**Synthesis of ¾.** In a glovebox, a flame-dried Schlenk reaction tube equipped with a magnetic stir bar, were added NHC precatalyst **C5** (0.01 mmol,), $K_2CO_3$ (16.6 mg, 0.12 mmol), oxidant DQ (49.0 mg, 0.12 mmol), **1** (0.10 mmol), **2** (0.15 mmol), and freshly distilled $CH_2Cl_2$ (1.0 mL). The reaction mixture was stirred at room temperature until the starting material **1** was completely consumed (12–24 h). The mixture was then filtered through a pad of Celite washed with $CH_2Cl_2$. After solvent was evaporated, the residue was purified by flash column chromatography to afford the desired product **3/4**.

## Data availability

For ¹H, ¹³C NMR and high-performance liquid chromatography spectra of compounds in this paper, see Supplementary Figs. 1–187. For details of the synthetic procedures, see Supplementary Notes. The supplementary crystallographic data for this paper could be obtained free of charge from The Cambridge Crystallographic Data Centre (**6**: CCDC 1880233) via www.ccdc.cam.ac.uk/data_request/cif.

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

## Acknowledgements

Generous financial support for this work is provided by: the National Natural Science Foundation of China (Nos. 21672121 and 21871160), the "Thousand Plan" Youth Program of China, the Tsinghua University, the Bayer Investigator fellow, the Fellowshio of Tsinghua-Peking Centre for Life Sciences (CLS).

## Author contributions

G.M.Y. conducted the main experiments; D.H.G. and D.M. prepared the several starting materials, including substrates. J.W. conceptualized and directed the project, and drafted the paper with the assistance from co-authors. All authors contributed to the discussions.

## Additional information

**Competing interests:** The authors declare no competing interests.

