## [Peer Review File · Nature Communications]

Reviewers' comments:

Reviewer #1 (Remarks to the Author):

In this manuscript, Wang and coworkers described an efficient carbene-catalyzed atroposelective synthesis of axially chiral biaryl amino alcohols via desymmetrization/kinetic resolution sequence. This strategy exhibits excellent substrate compatibility, efficiency and enantiocontrol. Mechanistic investigations clearly demonstrated that the desymmetrization step should be the main factor for the remarkable stereoselectivity. Subsequent kinetic resolution was proposed to further enhance the stereocontrol by removing the undesired enantiomer. The practicality was illustrated by the gram-scale reaction, versatile synthetic transformations and metal mediated asymmetric catalysis. Overall, this work provided a concise approach to access optically active NOBIN derivatives which are inaccessible by conventional protocols. The cooperative strategy is a very nice design for this project and Prof. Akiyama's pioneering work should be cited (J. Am. Chem. Soc. 2013, 135, 3964). Therefore, I recommend the publication of this work in Nature Communications after the solution of the following issues.

- (1) To achieve good results, the Cbz protecting group seems essential. How about other protecting groups (Boc, Bz, and so on)?
- (2) For the asymmetric preparation of BINAM, List (Angew. Chem. Int. Ed. 2013, 52, 9293) and Kürti's (J. Am. Chem. Soc. 2013, 135, 7414) work should be cited.
- (3) Figure 7, please confirm the ee value (99% in the main text but 98% in figure 7).
- (4) In "a-pyrone-aryls", the "a" should be presented in symbol style "alpha".
- (5) The 3rd line of optimization, a "full stop" should be added after "DQ as oxidant".
- (6) Figure 4 (a), "experiments shows" should be changed to "experiments show".
- (7) Figure 6, "cat. 7 (10 mol%)" but not "cat. 6 (10 mol%)".
- (8) General procedure for preparation of racemic samples (compound 3 and 4) should be included in SI.

Reviewer #2 (Remarks to the Author):

Wang and co-workers report in this manuscript a highly efficient NHC-catalyzed desymmetrization of achiral amino bisphenols to access novel NOBIN analogs in high yield and enantioselectivity. As pointed out by the authors, the access to enantiopure NOBINs still represents a challenge in asymmetric synthesis despite numerous attempts on this topic. The development of new methods, especially practical ones, should be significant achievement. Some derivatisation and application of these new NOBIN analogs were also included in this work. Overall, the experimental part of this work is of high quality, but in my opinion, it does not justify publication in a top and general journal such as Nature Communications. It would make a nice contribution to more organic chemistry-focused journals. My reasons are listed below:

1) The most important concept that is emphasized in this manuscript is the "design" of desymmetrization followed by kinetic resolution to enhance the enantiopurity of the products. However, this is not a new design at all. It has been noted in many precedents and in particular, Akiyama has reported a closely related desymmetrization of bis-phenols (J. Am. Chem. Soc. 2013, 135, 3964–3970.), which explained the same concept of desymmetrization followed by kinetic resolution using a similar scheme in details. This important precedent, unfortunately, is not cited in this manuscript.

2) NHC-catalyzed enantioselective acylation of phenols have been reported in a few occasions including the Zhao group, the Chi group as well as the Wang group themselves. This desymmetrization of amino bisphenols is actually very similar to some previous work in terms of catalyst and reagent development. This work is rather a very nice extension of some of the previous work.

3) It is nice that application of the new NOBIN analogs were included. However, it is curious why some of the really poor results of ATH of ketones should be included (18-66% ee, which were reported in er to make it look better). The low ee really did not justify well for the new compounds prepared in this study.

Reviewer #3 (Remarks to the Author):

Wang and co-workers describe an chiral NHC catalyzed desymmetrization reaction to prepare axially chiral biaryl amino alcohols. This is a rare case that chiral NHCs are applied to atroposelective reactions. The summarized results are certainly interesting and broadens the toolbox for the synthesis of chiral biaryl compounds, a class of privileged ligands for asymmetric synthesis. The substrate scope for the aniline moiety is particularly diversified. The authors demonstrate that a class of phenyl analogues of NOBIN can be prepared in excellent yield and ee. They further show that derivatives of those products can be used as chiral ligands for asymmetric alkylation and hydrogenation reactions. Publication is recommended pending minor revisions.

1. The catalysts used in this study are N-heterocyclic carbenes (NHC). It is more appropriate to use NHC rather than the general term of carbene.
2. Can the authors comment on why a remove nitro group on the catalyst exhibits superior enantioselectivity? A control catalyst, in which Ar = 2,4,6-triisopropylphenyl, X = H, should be tested to compare this nitro effect.
3. How much does the kinetic resolution step (the second acylation) really contribute to the overall selectivity? Yield is high for most substrates, suggesting little diester is formed. In Figure 4, Eq. 1, the desymmetrization step is highly selective ($K_{fast}/K_{slow} = 28:1$, BTW, ent- is confusing, it is suggested to use R and S to indicate stereochemistry). This means the concentration of the minor enantiomer of 3a is extremely low. On the other hand, the selectivity of the kinetic resolution is moderate. Therefore, more major enantiomer would be converted to the achiral diester due to concentration advantage. The result of higher ee is because the minor enantiomer is consumed eventually. This is a classical example of enhancing ee at the expense of yield, same as ref. 66.

Response to reviewer 1:

1. In this manuscript, Wang and coworkers described an efficient carbene-catalyzed atroposelective synthesis of axially chiral biaryl amino alcohols via desymmetrization/kinetic resolution sequence. This strategy exhibits excellent substrate compatibility, efficiency and enantiocontrol. Mechanistic investigations clearly demonstrated that the desymmetrization step should be the main factor for the remarkable stereoselectivity. Subsequent kinetic resolution was proposed to further enhance the stereocontrol by removing the undesired enantiomer. The practicality was illustrated by the gram-scale reaction, versatile synthetic transformations and metal mediated asymmetric catalysis. Overall, this work provided a concise approach to access optically active NOBIN derivatives which are inaccessible by conventional protocols. The cooperative strategy is a very nice design for this project and Prof. Akiyama's pioneering work should be cited (*J. Am. Chem. Soc.* 2013, 135, 3964). Therefore, I recommend the publication of this work in *Nature Communications* after the solution of the following issues.

Answer:

We thank this reviewer to give a positive response. And we have added this reference in the revised manuscript. Please see the new reference 42 (Mori, K., Ichikawa, Y., Kobayashi, M., Shibata, Y., Yamanaka, M., Akiyama, T. Enantioselective synthesis of multisubstituted biaryl skeleton by chiral phosphoric acid catalyzed desymmetrization/kinetic resolution sequence. *J. Am. Chem. Soc.* **135**, 3964-3970 (2013)).

2. To achieve good results, the Cbz protecting group seems essential. How about other protecting groups (Boc, Bz, and so on)?

Answer:

We have also examined several other protecting groups, such as Boc and Pht. Similar results as well as Cbz group have been achieved (For details, See Supplementary Table 2 in SI). Meanwhile, the Cbz group can be easily removed under neutral condition. Consequently, "Cbz" was selected finally as the protecting group in this protocol.

3. For the asymmetric preparation of BINAM, List (*Angew. Chem. Int. Ed.* 2013, 52, 9293) and Kürti's (*J. Am. Chem. Soc.* 2013, 135, 7414) work should be cited.

Answer:

We have added the references in the revised manuscript. Please see the new reference 13 (De, C. K., Pesciaioli, F., List, B. Catalytic asymmetric benzidine rearrangement. *Angew. Chem.* **125**, 9463-9465 (2013).) and reference 14 (Li, G. Q., Gao, H., Keene, C., Devonas, M., Ess, D. H., Kürti, L. Organocatalytic aryl-aryl bond formation: an atroposelective [3, 3]-rearrangement approach to BINAM derivatives. *J. Am. Chem. Soc.* **135**, 7414-7417 (2013)).

4. Figure 7, please confirm the ee value (99% in the main text but 98% in figure 7).

Answer:

After the double-check, the ee value of 3a in figure 7 has been confirmed as 99% and also corrected in the revised manuscript. Please check our revised manuscript.

5. In “a-pyrone-aryls”, the “a” should be presented in symbol style “alpha”.

Answer:

Followed this reviewer’s suggestion and we have correct it accordingly. Please check our revised manuscript (Page 1, left column).

6. The 3rd line of optimization, a “full stop” should be added after “DQ as oxidant”.

Answer:

We have corrected it accordingly. Please check it in the revised manuscript (Page 1, right column).

7. Figure 4 (a), “experiments shows” should be changed to “experiments show”.

Answer:

Followed this reviewer’s suggestion and we have correct it accordingly. Please check our revised manuscript in Figure 4.

8. Figure 6, “cat. 7 (10 mol%)” but not “cat. 6 (10 mol%)”.

Answer:

Followed this reviewer’s suggestion and we have correct it accordingly. Please check our revised manuscript.

9. General procedure for preparation of racemic samples (compound 3 and 4) should be included in SI.

Answer:

Followed this great suggestion and we have added the general procedure for the

preparation of racemic samples in SI. Please check it (See Supplementary Note 3).

Response to reviewer 2:

1. Wang and co-workers report in this manuscript a highly efficient NHC-catalyzed desymmetrization of achiral amino bisphenols to access novel NOBIN analogs in high yield and enantioselectivity. As pointed out by the authors, the access to enantiopure NOBINs still represents a challenge in asymmetric synthesis despite numerous attempts on this topic. The development of new methods, especially practical ones, should be significant achievement. Some derivatisation and application of these new NOBIN analogs were also included in this work. Overall, the experimental part of this work is of high quality, but in my opinion, it does not justify publication in a top and general journal such as Nature Communications. It would make a nice contribution to more organic chemistry-focused journals.

Answer:

Thanks this reviewer for the kind attention to review our manuscript. Why we believe our results can be published on Nat. Commun.? Several evidences listed below as reference:

- 1) Axially chiral biaryl amino alcohols (especially NOBIN) are recognized as one of the most useful molecules (e.g. chiral catalyst or ligand). However, the rapidly and highly atropenantioselective preparation of NOBIN type molecules is still highly desirable.
- 2) Our synthesized compounds belong to non-classic NOBIN-type derivatives (e.g. biphenyl- or phenyl-naphthyl-type amino-alcohols), offering extra possibilities as unusual ligand or catalyst to support asymmetric catalysis.
- 3) Most importantly, we here designed an unprecedented cooperative strategy (a desymmetrization followed with kinetic resolution) to complete the non-classic NOBIN synthesis. This new protocol can deliver axially chiral biaryl amino-alcohols with excellent enantioselectivities (most examples >95% ee).

Finally, we wish this reviewer to agree with our opinion.

2. The most important concept that is emphasized in this manuscript is the "design" of desymmetrization followed by kinetic resolution to enhance the enantiopurity of the products. However, this is not a new design at all. It has been noted in many precedents and in particular, Akiyama has reported a closely related desymmetrization of bis-phenols (J. Am. Chem. Soc. 2013, 135, 3964–3970.), which explained the same concept of desymmetrization followed by kinetic resolution using a similar scheme in details. This important precedent, unfortunately, is not cited in this manuscript.

Answer:

In the beginning, I have to apologize for mistake on missing Akiyama's work. Now, the work (Mori, K., Ichikawa, Y., Kobayashi, M., Shibata, Y., Yamanaka, M., Akiyama, T. Enantioselective synthesis of multisubstituted biaryl skeleton by chiral phosphoric acid catalyzed desymmetrization/kinetic resolution sequence. *J. Am. Chem. Soc.* **135**, 3964-3970 (2013)) has been added as reference 42.

In last a few years, our group already showed a big interest in exploring novel carbene-catalyzed transformations for the rapid assembling of axially chiral molecules. For examples, we have successfully reported the carbene-catalyzed atropenantioselective [3+3] annulation and kinetic resolution of anilides, respectively (see ref. 32-33 in revised manuscript).

In fact, there are several differences between the Akiyama's strategy and our design. For example, they worked in different catalyst system, and catalytic model (noncovalent catalysis vs covalent catalysis). Most importantly, there is no similar concept has been reported in NHC-organocatalysis field. We are the first group to utilize it to atropenantioselectively synthesize axially chiral biaryl amino alcohols.

3. NHC-catalyzed enantioselective acylation of phenols have been reported in a few occasions including the Zhao group, the Chi group as well as the Wang group themselves. This desymmetrization of amino bisphenols is actually very similar to some previous work in terms of catalyst and reagent development. This work is rather a very nice extension of some of the previous work.

Answer:

Thanks for this critical question. Indeed, the NHC-catalyzed enantioselective acylation of phenols has been broadly studied to synthesize chiral compounds (Note: ref. 31 is the only exception). Therefore, rapid synthesis of axially chiral biaryl amino-alcohols in highly atropenantioselective fashion is still in its infancy. In addition, our work is very different from the Zhao's and Chi's method. As showed in Zhao's work (Ref. 46), the long distance of two phenol groups leads to a low reactivity and poor enantioselectivity (60% yield, 23% ee). Some other successful examples demonstrated that the effective cooperation of the two OH groups may be the key factor for high enantio-control (See below Figures). To our knowledge, the NHC-catalyzed highly enantioselective desymmetrization of 1,3-diphenol is still a big challenge by far. Fortunately, our work partially solves this problem and gives a positive contribution to this field.

4. It is nice that application of the new NOBIN analogs were included. However, it is curious why some of the really poor results of ATH of ketones should be included (18-66% ee, which were reported in er to make it look better). The low ee really did not justify well for the new compounds prepared in this study.

Answer:

Thanks for your impressive and critical question. As showed in Figure 6 in the manuscript, the ligands **15-17** just represent three different axially chiral biaryl amino-alcohols derivatives with different levels of enantioselectivities. These results (18-66% ee) just tell us a story that substrate has a significant effect on stereo-control. Given a comprehensive investigation on substrate, a superior chiral ligand or catalyst for a specific reaction can be achieved in principal. Similar stories also indicated in the Chi's work (*Angew. Chem. Int. Ed.* **58**, 1784 (2019)) and the Tan's work (*Nat. Commun.* **10**, 566 (2019)). Therefore, we think these examples are effective and useful.

Response to reviewer 3:

1. Wang and co-workers describe an chiral NHC catalyzed desymmetrization reaction to prepare axially chiral biaryl amino alcohols. This is a rare case that chiral NHCs are applied to atroposelective reactions. The summarized results are certainly interesting and broadens the toolbox for the synthesis of chiral biaryl compounds, a class of privileged ligands for asymmetric synthesis. The substrate scope for the aniline moiety is particularly diversified. The authors demonstrate that a class of phenyl analogues of NOBIN can be prepared in excellent yield and ee. They further show that derivatives of those products can be used as chiral ligands for asymmetric alkylation and hydrogenation reactions. Publication is recommended pending minor revisions.

Answer:

Thanks the reviewer for this positive commence.

2. The catalysts used in this study are N-heterocyclic carbenes (NHC). It is more appropriate to use NHC rather than the general term of carbene.

Answer:

Followed this reviewer's great suggestion and we have corrected it accordingly. Please check our revised manuscript (All carbenes have been replaced by NHC).

3. Can the authors comment on why a remove nitro group on the catalyst exhibits superior enantioselectivity? A control catalyst, in which Ar = 2,4,6-triisopropylphenyl, X = H, should be tested to compare this nitro effect.

Answer:

Thanks for your great suggestion. The control catalyst, in which Ar = 2,4,6-triisopropylphenyl, X = H has been examined. For details of catalyst investigation, please see the SI (Table 2, P_{s98}, catalyst C8 (70% yield, 90% ee)). In comparison, the catalyst C5, in which Ar = 2,4,6-triisopropylphenyl, X = NO₂, is the best choice.

4. How much does the kinetic resolution step (the second acylation) really contribute to the overall selectivity? Yield is high for most substrates, suggesting little diester is formed. In Figure 4, Eq. 1, the desymmetrization step is highly selective ($K_{fast}/K_{slow} = 28:1$, BTW, ent- is confusing, it is suggested to use R and S to indicate stereochemistry). This means the concentration of the minor enantiomer of **3a** is extremely low. On the other hand, the selectivity of the kinetic resolution is moderate. Therefore, more major enantiomer would be converted to the achiral diester due to concentration advantage. The result of higher ee is because the minor enantiomer is consumed eventually. This is a classical example of enhancing ee at the expense of yield, same as ref. 66.

Answer:

Thanks for this reviewer's great suggestion. We now use optical rotation to isolate two enantiomers in Figure 4 (Note: optical rotations for all compounds have been placed in SI). (-)-**3a** means major enantiomer (Actually, it is (*R*)-configuration). (±)-**3a** means racemic **3a**. Meanwhile, we agreed with this reviewer's opinion that the desymmetrization provides the major contribution for high enantio-control based on experimental result ($K_{fast}/K_{slow} = 28:1$). For sure, another plausible reason for high ee caused by eventual consuming of minor enantiomer can't be completely ruled out.

REVIEWERS' COMMENTS:

Reviewer #3 (Remarks to the Author):

Most issues raised by the reviewers are addressed in the revised draft. Publication is recommended.